# Stromal Protein-Mediated Immune Regulation in Digestive Cancers

**DOI:** 10.3390/cancers13010146

**Published:** 2021-01-05

**Authors:** Pia Gamradt, Christelle De La Fouchardière, Ana Hennino

**Affiliations:** 1Cancer Research Center of Lyon, UMR INSERM 1052, CNRS 5286, F-69373 Lyon, France; christelle.delafouchardiere@lyon.unicancer.fr; 2Department of Medical Oncology, Léon Bérard Center, F-69008 Lyon, France; 3Université Lyon 1, F-69100 Lyon, France

**Keywords:** tumor microenvironment (TME), extracellular matrix (ECM) proteins, immune regulation, digestive cancers

## Abstract

**Simple Summary:**

Solid cancers are surrounded by a network of non-cancerous cells comprising different cell types, including fibroblasts, and acellular protein structures. This entire network is called the tumor microenvironment (TME) and it provides a physical barrier to the tumor shielding it from infiltrating immune cells, such as lymphocytes, or therapeutic agents. In addition, the TME has been shown to dampen efficient immune responses of infiltrated immune cells, which are key in eliminating cancer cells from the organism. In this review, we will discuss how TME proteins in particular are involved in this dampening effect, known as immunosuppression. We will focus on three different types of digestive cancers: pancreatic cancer, colorectal cancer, and gastric cancer. Moreover, we will discuss current therapeutic approaches using TME proteins as targets to reverse their immunosuppressive effects.

**Abstract:**

The stromal tumor microenvironment (TME) consists of immune cells, vascular and neural structures, cancer-associated fibroblasts (CAFs), as well as extracellular matrix (ECM), and favors immune escape mechanisms promoting the initiation and progression of digestive cancers. Numerous ECM proteins released by stromal and tumor cells are crucial in providing physical rigidity to the TME, though they are also key regulators of the immune response against cancer cells by interacting directly with immune cells or engaging with immune regulatory molecules. Here, we discuss current knowledge of stromal proteins in digestive cancers including pancreatic cancer, colorectal cancer, and gastric cancer, focusing on their functions in inhibiting tumor immunity and enabling drug resistance. Moreover, we will discuss the implication of stromal proteins as therapeutic targets to unleash efficient immunotherapy-based treatments.

## 1. Introduction

Colorectal, gastric, and pancreatic cancer represent three of the most frequent digestive cancers accounting for more than three million new cases per year worldwide as reported by the International Agency for Research on Cancer in 2018 [1]. Even though they share common characteristics, they mainly differ by their prognosis, being overall better for colorectal than gastric and pancreatic cancer. When metastatic, patient overall survival ranges from 12 to 30 months for colorectal cancer, depending on the patient’s rat sarcoma viral oncogene homolog (*RAS*) and B-Raf murine sarcoma viral oncogene homolog B1 (*BRAF*) statuses, and does not usually exceed 12 months for gastric and pancreatic cancer. Therapeutic management for digestive cancers is based on chemotherapy and if feasible, surgery of metastatic sites. Targeted therapies based on vascular endothelial growth factor (VEGF) receptor as well as human epidermal growth factor receptor (GFR) inhibition can be indicated in colorectal and gastric cancer on analysis of individual biomarkers [2,3,4,5]. Checkpoint inhibitor-based treatments are approved in cases harboring microsatellite instability or with high tumor mutational burden [6,7,8,9,10,11].

Colorectal, gastric, and pancreatic cancer are often characterized by an abundant desmoplastic stroma reaction that can outnumber the neoplastic cells by far and harbors the tumor microenvironment (TME). The stromal TME is composed of non-cancerous cells including immune cells (e.g., lymphocytes, macrophages, and myeloid-derived suppressor cells (MDSCs)), vascular and neural structures, as well as cancer-associated fibroblasts (CAFs) [12,13]. The cellular composition and differentiation of the TME is complex and has been extensively reviewed elsewhere [14,15,16]. In addition to the cellular compartment, the TME comprises a large amount of acellular components: the tumor-associated extracellular matrix (ECM), which consists of a complex meshwork of structural and regulatory proteins [17]. ECM proteins provide versatile signals to cells via the engagement with cell-surface receptors such as integrins or the binding and modulation of growth factors to activate pathways that are implicated in cellular proliferation, survival, adhesion, motility, and morphology [18,19]. Hence, ECM proteins of the TME have been recognized to contribute to cancer progression and invasion directly by promoting cellular transformation and metastasis but also indirectly by affecting stroma-regulated cellular processes such as inflammation and angiogenesis [20,21]. Moreover, ECM protein-mediated fibrosis compromises immune cell access and drug delivery, and it promotes resistance against cytotoxic therapies [22,23,24]. 

## 2. Extracellular Matrix Proteins

ECM proteins are endowed with both structural and regulatory functions. The genes encoding for ECM proteins and ECM-associated proteins are collectively known as the matrisome, which encompasses two main groups of genes: the core matrisome genes encoding collagens, glycoproteins, and proteoglycans, or matrisome-associated genes encoding ECM-affiliated proteins, ECM regulators, and secreted factors that are involved in the regulation or modulation of ECM functions [17,25]. Among the core matrisome proteins, collagens and the glycoproteins elastin, fibronectin, and laminins are considered to be the main fibrous ECM proteins determining the ECM structure [26]. Another group of glycoproteins comprised of matricellular proteins, including secreted protein acidic and rich in cysteine (SPARC) and tenascin, has regulatory functions but does not contribute significantly to the ECM structure [27]. Proteoglycans are found in the extracellular interstitial space and are associated with a wide range of functions implicated in the formation and physiochemical properties of the ECM [28]. 

Although cancer cells have the ability to release ECM proteins into the TME, stromal cells (e.g., fibroblasts) and invading immune cells remain the main source of ECM proteins [29,30]. However, the ECM protein signature between stromal and tumor cells differs as demonstrated by a recent study analyzing different stages of pancreatic cancer in pre-clinical mouse models and clinical samples. Indeed, Tian et al. showed that stromal cells predominantly produce ECM proteins belonging to the core matrisome, while cancer cells produce a wide range of both core matrisome and matrisome-associated proteins [31]. Moreover, whereas cancer cell-derived ECM proteins are largely associated with poor survival, ECM proteins exclusively produced by stromal cells are either positively or negatively correlated with survival [31,32]. Another study analyzed changes in the ECM signature in metastatic sites of high-grade serous ovarian cancer ranging from low to extensive disease. They determined a set of proteins associated with the pre-metastatic niche and poor prognosis [13]. It is now widely accepted that ECM proteins of the stromal TME have both pro- and anti-tumoral functions acting at different levels of tumor progression (e.g., epithelial-to-mesenchymal transition (EMT), proliferation, migration) [18,20,21]. However, a more detailed understanding of the acellular TME compartments could provide targets for selective therapeutic approaches [33]. Albeit there is long-standing knowledge of the immune modulatory functions of many ECM-associated proteins, namely cytokines and chemokines, such properties of core matrisome ECM proteins have only recently emerged [34,35]. Therefore, in this review we will focus on this latter category of ECM proteins and on their interactions with immune cells promoting immune escape and immune suppression during the initiation and progression of colorectal, gastric and pancreatic cancer, which are associated with abundant stromal reactions [36,37,38].

## 3. Immune Regulation by ECM Proteins in Digestive Cancers

It is important to emphasize that very few reports have been devoted to studying the pitfalls of the immune surveillance mechanisms that lead to tumor immune escape in the context of digestive cancers. The immune system plays a crucial role in preventing tumorigenesis. Moreover, the overall survival and therapeutic efficiency is strongly linked to the presence and distribution of immune cells within the TME. Recently, two retrospective studies evaluating pancreatic tumor infiltration by innate and adaptive immune cells found an association between prolonged survival and presence of CD4+ and CD8+ T cells as well as the NKT cells and their interaction with macrophages [39,40]. Indeed, the immune system recognizes and eliminates tumor cells by exerting a selective pressure on tumor cells attempting to escape immune surveillance. This latter is composed of three steps: elimination, equilibrium and escape, highlighting a coordinated effort by immune cell populations produced by the thymus and recruited locally to their target organ [41,42]. The elimination phase occurs at an early stage in tumor development. Natural killer (NK) cells of the innate immune system as well as adaptive immune cells, such as CD4+ and CD8+ αβ T cells and γδ T lymphocytes, are the unique actors of this phenomenon [42]. During this phase, neoplastic lesions are eliminated before any clinical signs occur. However, some neoplastic lesions may not be completely eliminated and acquire additional mutations, rendering them resistant to the cytotoxic attacks of the immune system. This phase is known as an equilibrium phase [43]. Very few reports have focused on this phase since it is clinically silent. However, during this phase, the immunogenicity of the tumor is remodeled by the pressure enforced by the immune system. During cancer development, the variants of the tumor cells escaping the CD8+ T cell cytotoxic attack proliferate. This immune evasion in cancer is associated with profound immune suppression in which the stroma plays an important role in dampening the effective immune response. Indeed, a recent study investigated the role of growth arrest-specific protein 6 (Gas6), a glycoprotein which is produced by macrophages and CAFs in the TME, in the suppression NK cells [44]. By employing antibody-mediated blockage of Gas6 in an orthotopic model of pancreatic cancer, the researchers could show that NK cell activation was restored, resulting in reduced tumor weight and less lung metastases [44]. Furthermore, the recruitment of MDSCs, tumor-associated macrophages (TAMs) and regulatory T (T reg) cells skews the immune response and modifies the pancreatic microenvironment favoring immune escape [45,46]. TAMs have been divided into M1 and M2 macrophages by analogy to helper T cell differentiation [47,48].

In the following section we will describe the implication of several core matrisome ECM proteins associated with colorectal, gastric, and pancreatic cancer in promoting immune modulation, which facilitates immune evasion of neoplastic cells (Table 1).

### 3.1. Collagens

Collagen is the most abundant ECM protein that self-assembles into fibrils, provides tissue support for cell growth, and contributes to mechanical resilience of connective tissue [49]. The broad diversity of collagens and their complex assembly has been extensively reviewed elsewhere [50,51,52]. Briefly, each collagen molecule consists of a triple helical region formed by three polypeptide chains, and two non-helical regions at either end. To date, 29 distinct homo- and/or heterotrimeric molecules have been described. The collagen molecules are wound into each other forming rope-like structures with a length and diameter of 280 nm and 1.4 nm, respectively. Collagen organization strongly influences cell behavior and tissue mechanics [53,54,55]. By using two-photon microscopy on tissue samples from patients with ovarian cancer to visualize collagen fibers using second harmonic generation label-free illumination, Pearce et al. showed that in tissues with a low disease score, collagen fibers are thin and mostly arranged around the adipocytes [13]. In contrast, in tissues with a high disease score denser arrays of long collagen bundles with a predominant micro-scale orientation were observed, and this orientation was largely correlated with disease score [13]. As with the disease score, the collagen composition appeared to change over time as shown by another study comparing the ECM composition in different states of pancreatic cancer as well as pancreatitis [31]. While the fibrillar collagens COLA1A1, COL1A2, and COL3A1 are the most abundant in all stages of pancreatic cancer and pancreatitis in both mouse and human, COL6A3 is highly enriched in advanced pancreatic ductal adenocarcinoma (PDAC) [31]. Moreover, the representation of collagens (as well as other ECM proteins) not only differs between heathy and tumor tissue but also between the initial tumor and the metastatic site [56,57].

As collagen is the most abundant ECM protein, it seems reasonable to assume that it does not only provide mechanical functions but also affects invading immune cells. Mechanically collagen provides tissue stiffness and rigidity to the TME, which can lead to the exclusion of immune cells such as CD8+ T cells from the tumor [58,59]. Interestingly, a study analyzing the T cell infiltration of pancreatic cancer showed that although the spatial distribution of T cells within in the tumor is relevant for patient outcome, ECM collagen has no significant impact on their infiltration [60], similarly to T cell infiltration in breast cancer [61]. However, in this particular study the activation status of infiltrated T cells was not determined [60]. Nevertheless, it has been shown that high collagen density can reduce the cytotoxic activity of tumor-infiltrating T cells and up-regulate regulatory cell markers [62]. 

Interestingly, it has been shown that experimental tumors combining Kristen rat sarcoma gene (*KRAS*) mutations and transforming growth factor (TGF)-β type II receptor deletion are highly fibrotic and exhibit a pronounced mesenchymal-like phenotype following stromal ablation [63,64]. Laklai et al. showed that in PDAC patient biopsies, higher collagen I content and activated STAT3 were associated with SMAD4 mutation and shorter survival [65]. These findings suggest the implication of epithelial tension and matricellular fibrosis in the aggressiveness of SMAD4 mutant pancreatic tumors, and highlight greater CD68^+^ macrophage recruitment to stiffer STAT3+ zones as key drivers of this phenotype.

To further investigate the immune modulatory function of collagen, Pinto and colleagues employed a model of decellularized tissue samples from colorectal cancer patients, in which they showed that collagen was the main remaining component of the scaffold following decellularization [66]. Moreover, the decellularized ECM scaffold promoted macrophage polarization toward the anti-inflammatory M2 phenotype, as indicated by their release of interleukin (IL)-10, TGF-β, and CCL18 [66]. As such, educated M2 macrophages also drove cancer cell invasion in a CCL18-dependent manner [66]. A recent study on gastric cancer revealed that the tumor immune cell infiltrate was correlated with the expression of different collagens [67]. This expression was in turn associated with the presence of M2 macrophages, indicating that they promote immunosuppressive phenotypes [68].

### 3.2. Fibrous and Non-Fibrous Glycoproteins

#### 3.2.1. Fibronectin

Fibronectin is a high molecular weight glycoprotein consisting of two subunits that range in size (230–270 kDa) depending on alternative splicing and that are linked by disulfide bonds between c-terminal cysteine residues of each subunit [69]. Fibronectin exists as soluble plasma-fibronectin that is secreted by hepatocytes into the bloodstream [70] or as a structural protein expressed in tissues by fibroblasts and other cell types [71,72]. Fibronectin mainly interacts through integrins, and due to alternative splicing and posttranslational modulations several isoforms of fibronectin exist. Fibronectin mediates multiple cellular processes such as attachment, motility, contractility, and ECM assembly [69,71,72]. Fibronectin binds to several ECM proteins including collagen, which leads to ECM maturation implicated in both physiological processes and pathologies [73]. 

As mentioned above, tumors are infiltrated by TAMs derived from circulating monocytes that can be pro- or anti-inflammatory depending on their polarization. Fibronectin produced by the TME appears to be crucial for the recruitment of TAMs, since it has been demonstrated to facilitate the migration of mouse and human macrophages by activating the focal adhesion kinase (FAK) pathway [74]. Here, FAK activation, which was shown to be mediated by the interaction of fibronectin and integrin α5β1 [75], resulted in colony-stimulating factor 1 receptor (CSF-R1) phosphorylation required for macrophage migration [74]. Moreover, the interaction of monocytes with fibronectin modulates their expression of matrix metalloproteinases (MMPs), especially MMP9, when they are co-cultured with gastric carcinoma-derived cancer cells, whereas depletion of fibronectin results in diminished MMP9 production [76]. However, in vitro cell invasion of gastric carcinoma-derived cancer cells was only promoted when the TAM-like monocytes expressed MMP9. In a follow-up study the same researchers demonstrated that the invasiveness of gastric carcinoma-derived cancer cells was promoted by tumor necrosis factor (TNF)-α released from macrophages in a fibrinogen-dependent manner [76,77]. 

Fibronectin not only promotes the invasiveness of the primary tumor but it also participates in preparing pre-metastatic niches. By employing an experimental model of intra-splenic PDAC cell injection, Costa-Silva and colleagues demonstrated that PDAC-derived exosomes fuse with hepatic Kupffer cells. This fusion results in TGF-β signaling-induced up-regulation of fibronectin by hepatic stellate cells, which was required for recruitment of bone marrow-derived macrophages [78]. Moreover, they showed that the pro-metastatic effect in this model was dependent on macrophage recruitment, since the elimination of CD11b+ cells using a diphtheria toxin-based approach resulted in fewer liver metastases [78]. 

Although the presented studies did not further investigate the impact of fibronectin on skewing the phenotype of TAMs toward M1 or M2, in other malignancies fibronectin has been demonstrated to affect the phenotype of infiltrating cells. As mentioned above fibronectin can be present in several isoforms depending on alternative splicing [79]. The alternatively spliced extra domain (ED) A of fibronectin is found in large amounts in the stroma of most solid tumors [80]. A study assessing the role of fibronectin in a mouse model of fibrosis showed that EDA-fibronectin acts on integrin α5β1 and converts myeloid cells into anti-inflammatory MDSCs with increased expression of arginase-1 [81]. While EDA-fibronectin pre-exposition of MDSCs is beneficial in fibrosis owing to the prevention of exacerbated inflammatory tissue damage, it is detrimental in mice injected with B16 melanoma cells as it results in arginase-mediated cancer growth [81]. In addition to recruiting and priming immunosuppressive cells, fibronectin controls the migration of tumor-infiltrating lymphocytes preventing them from encountering tumor cells. By analyzing the trajectories of T cells in freshly isolated human lung tumors, both CD4+ and CD8+ T cells were shown to be entrapped within fibronectin- and collagen-rich stromal regions and the restricted access to the tumor could be lifted by treating the tumor samples with collagenase [59]. These results imply that in addition to promote macrophage-derived MMP-mediated tumor invasion, fibronectin also contributes to immune evasion by hampering proper tumor infiltration by T cells. 

#### 3.2.2. Laminin

Laminins are a group of large molecular weight glycoproteins (~400 kDA), key in shaping the basement membrane that separates the endothelium and epithelium from underlying connective tissue [82,83]. As with fibronectin, laminins form networks by binding cellular receptors [84,85]. Currently, 16 laminin isoforms are known, constituted by the assembly of three disulfide-linked polypeptides, the α, β, and γ chain [84,85]. The location of laminins within the basement membrane, which is also a stem cell niche, advocates for their strong involvement in shaping the stem cell phenotype [86,87]. Although modulation of immune cell recruitment and functions by laminins has been described in addition to their structural function [88], their immunomodulatory role in digestive cancers remains to be clarified. As mentioned above, T cells can be excluded from solid tumors [59,89] and must cross the basement membrane to enter tumor areas. Interestingly, different laminins have been associated with permissive (LAMA8, LAMA4) [90,91] or non-permissive (LAMA5) T cell migration signals [92]. In a pre-clinical model of colorectal cancer, it was demonstrated that up-regulation of superoxide dismutase 3 (SOD3) enhances tumor infiltration by CD8+ T cells by activating WNT pathway in the ECM [93]. WNT activation normalized the tumor endothelium by up-regulating the expression of LAMA4 and becoming permissive to T cell transfer [93]. Moreover, tumor-infiltrating DCs, which are important to establish efficient T cell responses, have been shown to interact with laminins and to be affected by this interaction [94]. DCs isolated from tumors in a mouse model of ovarian cancer express CD49f, a well-known receptor of laminin [95,96]. DCs can establish long-term interactions with laminins resulting in impaired immunological functions. Such long-term in vitro culture of DCs in contact with laminin resulted in DCs less efficient in inducing T cell proliferation compared to DCs cultured in laminin-free control conditions [96]. Since dysregulated DC responses have been detected in digestive cancers [97], it is tempting to speculate that similar mechanisms of laminin-induced DC tolerance may arise in which the TME promotes tumor immune escape. 

#### 3.2.3. Elastin

While collagen is the structural ECM protein that provides support and resistance to tissues, elastin contributes to the elasticity of many soft tissues including large arteries, skin, lung, elastic cartilage, ligament and tendon [98]. Elastin is an insoluble polymer that is formed by extensive crosslinking of its precursor monomer tropoelastin [99]. Tropoelastin, which has a molecular weight of 60–70 kDA, is secreted by various cell types such as fibroblasts, endothelial, epithelial cells or smooth muscle cells [100]. Elastin degradation by elastases including serine-, cysteine-, and metalloproteinases is a process associated with aging but also malignancies such as cancer [101]. In a recent study, researchers showed that elastin expression increased in colorectal cancer and this increase was associated with high MMP9 and MMP12 expression, as well as that of their inhibitor TIMP 3 [102]. Moreover, they determined that TNF secretion by macrophages was induced by the interaction of macrophages and elastin. Thus, elastin contributes to the generation of a pro-tumorigenic microenvironment [102]. Elastin-associated inflammatory processes are usually mediated by fragments of the proteins, which arise during its degradation e.g., by metalloproteases. In line with the findings described above, another study reported the chemotactic activity of elastin fragments on monocytes in a mouse model of lung emphysema [103]. 

### 3.3. Matricellular Glycoproteins

Matricellular proteins are important components of the ECM, as they are endowed with pleiotropic regulatory but not structural functions involved in tissue remodeling and repair. These include proteins of the connective tissue growth factor, cysteine-rich proteins and the nephroblastoma overexpressed gene (CNN) family and secreted protein acidic and rich in cysteine (SPARC) family as well as thrombospondins, periostin, tenascins, and osteopontin. Matricellular proteins have been associated with many malignancies [104], and are produced by stromal and tumor cells but also by TME-invading cells such as macrophages [105]. They can bind to other structural ECM proteins such as collagens and fibronectin, influencing matrix formation and thereby promoting tumor growth. Moreover, matricellular proteins have been shown to interact with growth factors and integrins to regulate cell growth, adhesion, and motility [106]. As with other ECM proteins, matricellular proteins are substrates for metalloproteases and other proteases, and their cleavage can result in the exposure or rupture of active domains [107,108]. Moreover, matricellular proteins impact various cellular processes including immune responses through their matrix modulatory capacities, their interactions with cell-surface receptors and their cleavage by proteases [106]. 

#### 3.3.1. SPARC/Osteonectin

Secreted protein acidic and rich in cysteine (SPARC) (also referred to as osteonectin or basement membrane protein 40) is a 32 kDA matricellular protein involved in cell-cell interactions, growth factor signaling, and ECM remodeling via its regulation of collagen deposition and MMP production and activation [109,110,111,112,113,114]. In addition, SPARC has been reported to influence numerous pathways associated with proliferation, migration, differentiation, adhesion, apoptosis, wound healing and EMT [109,115,116,117,118]. Given its vast variety of functions, SPARC has been reported to be a tumor promoter [119,120,121] and suppressor [122,123] depending on the type of cancer. Interestingly, in pancreatic cancer a compartmentalization of SPARC can be observed and stromal SPARC expression is associated with poor patient survival independently of the tumor-derived SPARC [124]. To evaluate the effect of host-derived endogenous or tumor cell-derived SPARC, researchers performed subcutaneous injections of pancreatic cancer cells into SPARC-/- mice. SPARC displayed anti-tumorigenic properties by regulating ECM deposition and inhibiting the growth of tumor cells [125]. A similar effect of SPARC was unveiled in gastric cancer, where SPARC overexpression was associated with fewer lung metastasis [126]. Moreover, an altered distribution of macrophages was detected in SPARC-/- compared to WT control mice after pancreatic tumor cell injection. While WT mice displayed a marginal location of macrophages, an clear intratumoral distribution of macrophages was observed in SPARC-/- mice [125]. In a follow-up study, employing an orthotopic mouse model of pancreatic cancer, the researchers further characterized infiltrating macrophages. They found that overall infiltration increased in mice lacking SPARC expression. While the activation of M1 macrophages did not differ between SPARC-/- and WT mice, a significantly increase in M2 macrophage activation was observed in mice lacking SPARC [127]. Stabilin-1 has been described as a SPARC receptor on macrophages, which is internalized and subjected to endocytosis after binding. By permitting the internalization of SPARC, stabilin-1 was surmised to fulfill regulatory functions in macrophages driving their phenotype [128]. Moreover, SPARC-/- mice exhibited increased levels of activated TGF-β, which was accompanied with increased frequencies of T reg cells. While macrophage infiltration was unaffected and M2 activation was mildly increased upon TGF-β inactivation in SPARC-/- mice, the increase in T reg cells was abrogated resulting in reduced tumor progression and survival [129]. The authors speculated that alterations in the ECM due to the lack of SPARC increased TGF-β availability. Aside from these finding, the use of SPARC-/- mice must be carefully interpreted, since SPARC expression impacts both spleen development and lymphocyte trafficking through the spleen [130]. Also, SPARC-/- mice were shown to mount insufficient innate immune responses [130]. In addition to tumor cells and fibroblasts, M2 macrophages have been shown to be a major source of SPARC in gastric cancer [131]. Moreover, this recent study demonstrated that SPARC overexpression in M2 macrophages reduced M2-mediated functions including proliferation of gastric cancer cells, emphasizing the anti-tumorigenic role of SPARC in digestive cancers [131]. 

Also in other malignancies SPARC was reported to affect anti-tumor immune responses. A study using an oncogene-induced model of bladder cancer in SPARC WT and SPARC-/- mice, demonstrated that the loss of SPARC promotes the inflammatory phenotype of TAMs (as well as CAFs) through the activation of the transcription factors NF-κB and AP-1. Conversely, inhibition of the NF-κB and AP-1 in the presence of SPARC results in reduced expression of TGF-β1 and stromal cell-derived factor 1 (SDF1). Although here the loss of SPARC results in increased inflammation, the investigators observed increased tumor progression and decreased survival [132]. 

#### 3.3.2. Osteopontin

Osteopontin, which was first discovered in osteoblasts, where it is crucial for bone homeostasis, is a 34–62 kDA multifunctional secretory acidic glycoprotein [133]. Osteopontin is encoded by the secreted phosphoprotein 1 (SPP1) gene, which has five isoforms and is also associated with neoplastic transformations [134]. Osteopontin is not only produced within the ECM but also by immune cells including activated lymphocytes, macrophages and dendritic cells (DCs) and is therefore also called early T-lymphocyte activation-1 (Eta-1) [135]. Osteopontin is involved in a variety of cellular processes such as adhesion, motility, and survival and its functional versatility can be explained by its capacity to engage with multiple receptors including various integrins as well as CD44 variants [136]. While osteopontin has a physiological regulatory role in developmental processes and tissue remodeling, its aberrant expression is associated with cardiovascular diseases, diabetes, and cancer [137]. Osteopontin is the physiological ligand of CD44, which is expressed by a variety of cell types including activated T cells [138]. Binding of osteopontin to CD44 leads to the suppression of CD8+ T cell activation and proliferation [136]. Osteopontin production is regulated by interferon regulatory factor (IRF) 8, which is expressed in myeloid and cancer cells [139]. Interestingly, in colorectal cancer IRF8 is often silenced by DNA methylation [140]. Klement and colleagues showed that this loss resulted in elevated osteopontin levels leading to CD8+ T cell suppression and poor prognosis [141]. Aside from directly interacting with T cells, osteopontin also affects the recruitment and differentiation of macrophages in gastric cancer, promoting a pro-tumorigenic immune response. A study investigating the relationship between osteopontin expression and macrophages in human gastric cancer showed that osteopontin serves as a chemoattractant for macrophages and promotes their skewing towards a CD204+ M2 phenotype [142]. Consequently, high osteopontin expression in combination with high CD204 expression on macrophages is associated with poorer survival of the patients [142]. In line with these findings, another study based on a mouse model of gastric cancer induced by *Helicobacter pylori* infection showed that osteopontin deficiency suppresses the migration of macrophages, resulting in reduced proliferation of gastric cancer cells [143]. 

Immune regulatory functions of osteopontin have been demonstrated in several other neoplastic malignancies. Indeed, a study using a human lung adenocarcinoma cell line revealed that tumor cell-derived osteopontin regulates M2 polarization by the up-regulation of programmed cell death-ligand 1 (PD-L1) on macrophages, and osteopontin knockdown results in macrophages with increased CD4+ T cell activation capacity [144].

#### 3.3.3. Periostin

Periostin is a multimodular protein with four isoforms ranging in molecular weight from 83–93 KD [145]. Periostin is physiologically expressed in the periosteum, a specialized membrane covering the outer surface of bone [146], and other connective tissues rich in collagen, where it plays a key role in ECM structuring via interactions with collagen and several other ECM proteins [147,148,149].

In pancreatic cancer, periostin has been detected in cancer epithelial cells, pancreatic stellate cells, and tumor stroma [150,151,152]. High periostin expression is observed at advanced stages of the disease and is associated with poor survival [150,152,153]. In a hepatic metastasis model of pancreatic cancer, periostin was shown to be induced by granulin secreted from metastasis-associated macrophages (MAMs) [154]. Although the exact mechanism underlying hepatic stellate cell activation by MAMs to produce periostin are currently unknown, periostin seems to be essential for preparing the metastatic niche. In a colitis-associated colorectal cancer model using periostin-/- mice, lack of periostin expression resulted in fewer inflammatory peritoneal macrophages, as evidenced by a reduced expression of TNF-α and IL-1β, and up-regulation of TGF-β and IL-10 [155]. Interestingly, in patients with intestinal type gastric cancer, low periostin levels were associated with poor survival and lymph node metastasis [156]. The same study revealed that high periostin levels in diffuse gastric cancer was associated with high M2 macrophage infiltration, whereas this was not the case in intestinal gastric cancer. 

An interaction between periostin and TAMs has also been observed in several other malignancies. In ovarian cancer, the expression of periostin promotes tumor infiltration with macrophages. Moreover, periostin expression by ovarian cancer cells was induced by TGF-β from infiltrating macrophages, which in turn promoted their skewing towards the M2 phenotype [157]. In wound healing and scar formation, periostin-mediated up-regulation of TNF-α on infiltrating monocytes appears to be beneficial since it promotes the proliferation of pericytes and thereby enhances functional recovery after injury [158]. Finally, in a murine model of melanoma within an inflammatory environment as well as in the skin of melanoma patients the recruitment of M2 macrophages was promoted in the presence of high periostin levels [159]. 

#### 3.3.4. βig-h3/TGF-β-Induced Protein

βig-h3 (also known as TGF-β-induced protein or TGFβi) is a 68 kDa ECM protein containing a secretory signal sequence, a N-terminal cysteine-rich domain, four fasciclin 1 domains, and a RGD (Arg-Gly-Asp) cell adhesive motif that was first isolated from human lung adenocarcinoma treated with TGF-β [160,161]. βig-h3, is expressed in a wide variety of tissues and the physiological functions of βig-h3 include cell-matrix interactions and cell migration [162]. In addition, βig-h3 interacts with various integrins [163,164,165] as well as other ECM molecules such as type I, II, and IV collagens, fibronectin, the proteoglycans biglycan and decorin, as well as periostin [166,167]. 

We and others have recently demonstrated that βig-h3 plays an important role in modulating the tissue stiffness of the TME in pancreatic cancer and its expression is associated with poor outcome [168,169]. Moreover, we showed that βig-h3 is a key regulator of immune responses during pancreatic cancer where it is expressed by tumor and stromal cells. By employing pre-clinical mouse models of pancreatic cancer, we showed that βig-h3 inhibits CD8+ T cell proliferation and activation. Moreover, βig-h3 diminishes the production of IFNγ and TNF-α by macrophages skewing them toward a non-inflammatory phenotype. Interactions of βig-h3 with both CD8+ T cells and macrophages were facilitated by binding of CD61 on their surface [168].

Also, in other diseases, βig-h3 has been associated with poor outcome. We showed that this matricellular protein has immunosuppressive properties in an autoimmune context [170]. Also, in their recent study Fico and Santamaria-Martínez demonstrated that βig-h3 promotes breast cancer metastasis and plays an important role in tumor angiogenesis [171]. Moreover, employing a KO model of βig-h3 as well as in silico analysis of human breast cancer samples, they showed that βig-h3 overexpression is positively linked with tumor hypoxia and M2 macrophages but negatively associated with CD8+ T cell infiltration [171]. In fibrosis and wound healing, macrophage-derived βig-h3 that is released after the uptake of apoptotic cells is considered to control collagen turnover as shown in a study from Nacu et al. [172]. Here, βig-h3 controlled the expression of MMP14, which is a critical factor for physiological collagen turnover and prevention of fibrosis [172]. Recently, it has been demonstrated that βig-h3 (as well as tenascin C and fibronectin) is produced by M2 but not M1 TAMs in ovarian cancer, where it contributes to tumor migration and thereby might promote metastatic spreading [173]. 

### 3.4. Proteoglycans

#### Versican

Versican is a hyaluronan-binding proteoglycan with a molecular weight of ~400 kDA in its largest isoform V0. Four more isoforms V1, V2, V3, and V4 have been described that are generated by alternative splicing [28]. Versican is involved in cell adhesion, proliferation, migration, inflammation, and ECM assembly [174,175]. As with other ECM proteins, versican has pro- and anti-inflammatory properties, making it a key player in cancer development. 

Such a dual function was reported in a study investigating the role of cancer cell-derived versican in colorectal versus breast cancer cells [176]. On the one hand, colorectal cancer cell-derived versican promoted the polarization of macrophages toward the pro-inflammatory M1 phenotype with up-regulated release of IL-6, IL-12 and TNF-α. On the other hand, breast cancer cell-derived versican promoted a M2 phenotype [176]. Moreover, versican binding to TLR2 on myeloid cells promotes the production of IL-6 and TNF-α [177]. While versican in its intact form mostly promotes tolerogenic actions, versican proteolysis results in the release of the versikine, a versican-derived matrikine that antagonizes the tolerogenic actions of its parent by promoting immunogenicity [178]. In colorectal cancer, versikine has been shown to promote T cell infiltration through the regulation of CD103+ conventional DC differentiation [179]. In both DCs and macrophages, induction of IRF8 expression was promoted by versikine, resulting in increased T cell recruitment and activation [178,179]. In addition, in gastric cancer versican expression serves as a biomarker [180]. The analysis of human gastric cancer tissue revealed a positive correlation between versican expression and T reg cells as well as other immunosuppressive proteins such as TIGIT or IDO1 but not PD1 [181]

Such diverse actions of versican can also be observed in other diseases. In cervical cancer, high stromal versican expression is associated with low numbers of intratumoral CD8+ T cells [182]. In mesothelioma, tumor cell-derived versican promotes both macrophage migration and their polarization toward the M2 phenotype [183]. As with gastric cancer, versican expression was associated with the accumulation of T reg cells, which further contributes to mesothelioma progression by hampering the anti-tumor immune responses [183]. In a murine melanoma model, the expression of inhibitory checkpoint ligand PD-L1 on macrophages was reported to be dependent on TNF-α expression by macrophages. Indeed, tumor cell-derived versican regulated TNF-α production in a TLR2-dependent manner [184].

## 4. Stromal ECM Proteins as Selective Therapeutic Targets

Since CAFs have been shown to be the major source of ECM proteins in the TME [208], therapeutic approaches aimed at non-selectively depleting tumor-associated stromal cells using inhibitors of the hedgehog pathway, which resulted in improved outcome in pre-clinical mouse models [24]. Then, hedgehog inhibitors have been tested in patients with pancreatic and colorectal cancer but their administration was interrupted since they failed to reproduce promising pre-clinical results in a clinical setting and even paradoxically accelerated diseases progression [209,210,211,212,213,214]. These findings provided insight into the constitution of TME that comprises components that can either promote or restrain tumor progression, and that these components need to be targeted in a selective manner in order develop novel therapeutic agents [33]. Indeed, several pre-clinical and clinical studies addressed various ECM proteins as targets. In the following section, we will discuss several therapeutic approaches tested (Figure 1).

### 4.1. Hyaluronan and Hyaluronidases

Hyaluronan (HA) is an extracellular glycosaminoglycan component that binds to ECM proteins contributing to their complex network. HA accumulation was observed in many solid cancers including PDAC where it is associated with more aggressive disease progression and limited delivery and distribution of therapeutic agents [215,216]. Since HA accumulation is associated with the deposition of collagen and proteoglycans forming a pro-tumorigenic TME, it represents a very attractive therapeutic target to make the TME more permissive to invading immune cells and therapeutic agents [217,218,219]. Hyaluronidases are enzymes with the capacity to depolymerize HA, and were formerly used in oncology to facilitate the access of chemotherapeutic agents to the tumor. Furthermore, depletion of HA through hyaluronidases diminishes collagen synthesis and thereby remodels the TME [220,221]. However, clinical development of hyaluronidases was limited due to poor pharmaceutical characteristics including allergic reactions and development of antibodies again bovine-derived hyaluronidase [222]. Recombinant human hyaluronidases have been developed and recently, pegvorhyaluronidase alfa (PEGPH20), a pegylated hyaluronidase, was developed in the hope of reducing systemic clearance and prolonging circulatory time of the molecule. In a clinical trial investigating the efficiency of PEGPH20 in PDAC, the pegylated hyaluronidase was combined with nab-paclitaxel/gemcitabine (PAG). As a control nab-paclitaxel/gemcitabine (AG) treatment was applied [223]. The results of the randomized phase II study advocated for the use of the PAG over AG regimen, although the delay in median progression-free survival (PFS) was modest (0.7 months) [223]. Moreover, the PFS increased within the subgroup of patients with high HA-expressing tumors that were defined as displaying ≥50% of hyaluronan positive staining within the TME. Although a pre-clinical study investigating the effect of PEGPH20 in a mouse model of pancreatic cancer showed that HA depletion resulted in increased intratumoral effector T cells [224], these results were not confirmed in a follow-up phase III trial [225]. Indeed, the PAG regimen did not improve the overall survival even though the selected patient cohort had been restricted to HA high-expressing tumors [225]. In addition, a genetically modified adenovirus encoding human PH20 hyaluronidase (VCN-01) was evaluated for its potential intratumoral administration to PDAC patients in combination with nab-paclitaxel/gemcitabine treatment. To date, the results have not been disclosed (NCT02045589). 

### 4.2. Immunocytokines

The extradomain B of fibronectin, which is selectively expressed in pancreatic cancer cells, was evaluated as a potential target for the immunocytokine L19-IL2, delivering IL-2 in a specific antibody-mediated manner [226]. A phase I clinical study was initiated combining escalating doses of L19-IL2 and gemcitabine in patients with advanced pancreatic cancer (NCT01198522). However, the study was terminated untimely due to poor patient recruitment. 

T cells derived from human peripheral blood mononuclear cells (PBMCs) can be stimulated with a recombinant fragment of fibronectin (FN-CH-296) containing the ligand for VLA-4 and -5, resulting in enhanced T cell effector functions and memory formation [227]. Based on these results, a phase I clinical trial (UMIN000001835) with transfer of FN-CH296 stimulated T cells was conducted in a cohort of nine patients with confirmed esophageal, gastric, colorectal, pancreatic, biliary tract or non-small lung cancer. Transfer of activated T cells resulted in a response rate of 22.2% and a disease control rate of 66.7% [228]. 

### 4.3. Inhibition of FAK Activity

FAK is often overexpressed and activated in advanced solid tumors such as PDAC. Since it promotes an immunosuppressive TME, FAK has emerged as a potential therapeutic target [229,230]. Indeed, Zaghdoudi and colleagues showed in a pre-clinical model of pancreatic cancer that FAK inhibition within fibroblasts resulted in increased sensitivity to chemotherapy and immune checkpoint inhibitor blockade. Moreover, they observed a decrease in the occurrence of metastases [231]. Currently, targeting FAK activity is evaluated in several phase I-II clinical trials as monotreatment, as well as in combination with programmed cell death protein 1 (NCT02758587) or mitogen-activated protein kinase kinase (MEK) inhibitors (NCT02428270). 

### 4.4. Inhibition of Matrix Metalloproteases

MMPs are family of 28 proteolytic enzymes with the capacity to degrade numerous components of the ECM and thereby controlling its turnover [232]. High levels of MMPs have been associated with prognosis in multiple cancers [233]. MMPs enhance tumor progression by promoting local invasion through the degradation of molecular components of basement membranes, tumor stroma, or vascular basal laminas [234]. This contribution to tumor progression as well as promising results in pre-clinical cancer models, made MMPs an attractive therapeutic target and several of them have been evaluated; however, with minor success [235]. For instance, a phase I study of COL-3, an oral MMP inhibitor, in patients with refractory metastatic cancer (NSC-683551) or a dose-seeking trial of PCK3145, an MMP-9 inhibitor, in metastatic prostate cancer (NCT00695851) have been closed without continuation. More recently, a phase I/IIa trial of BT1718 a “bicycle drug conjugate”, which inhibits the function of membrane type 1 (MT1)-MMP has been enrolled (NCT03486730). This agent is supposed to have a double function, since in addition to inhibiting MT1-MMP, after the attachment to its target a segment of the agent (DM1 toxin) in taken into the cancer cell causing it to undergo cell death. However, not results have been available to date.

### 4.5. Albumin-Based Agent Delivery

Nab-Paclitaxel, a 130 nm albumin-bound formulation of paclitaxel particles, was shown to exert antitumor activity in various cancers that overexpress SPARC, including breast cancer [236,237], lung cancer [238,239], and melanoma [240,241]. Therefore, SPARC positivity within the PDAC microenvironment was initially hypothesized to increase the concentration of nab-paclitaxel in tumors, as well as to serve as a predictive biomarker for nab-paclitaxel efficacy [242]. A phase I/II clinical trial revealed substantial antitumor activity after the administration of nab-paclitaxel in combination with gemcitabine [242]. However, in a phase III trial assessing the administration of nab-paclitaxel in combination with or without gemcitabine, no association between SPARC expression and overall survival could be determined [243].

## 5. Conclusions

This review provides an overview of the immune modulatory functions of stromal ECM proteins in digestive cancers contributing to the immune evasion of cancer cells. Moreover, we addressed treatment regimens that are currently under evaluation and interfere with immune modulation by targeting main components of the TME. In light of the importance of strong immune responses in successfully treating digestive cancer, a better understanding of how ECM proteins affect this response and could subsequently be targeted is urgently needed. We are certain that lessons can be learned from other solid tumors in which stromal immune modulation also occurs. However, future research in the field of digestive cancers should integrate the early interactions between stromal proteins and the outcome of the immune response in relevant GEMM mouse models. Understanding the mechanisms driving and modulating the orientation of the effective immune responses as a part of the evolving and progressing tumorigenesis is key to propose adapted and personalized therapeutics helping to unleash antitumor immune response. 

## Figures and Tables

**Figure 1 cancers-13-00146-f001:**
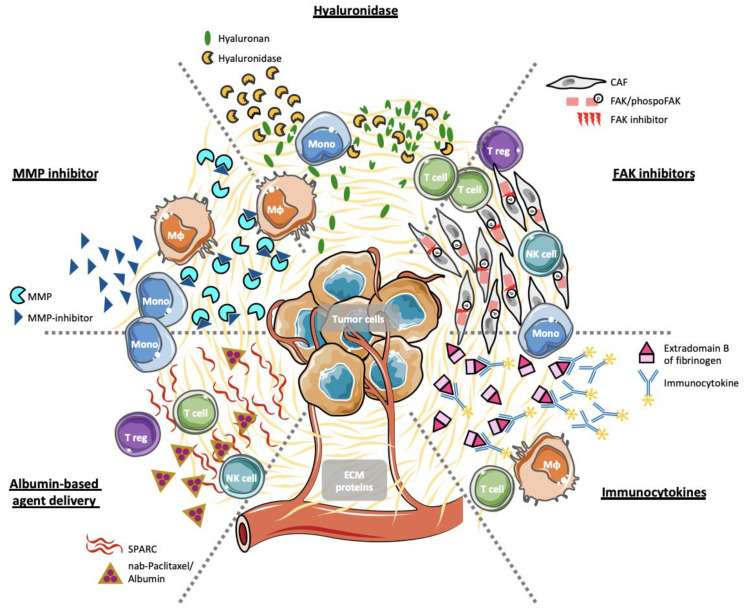
Therapeutic approaches targeting ECM proteins. ECM proteins within the TME have been selectively targeted by different approaches: Albumin-bound nab-Paclitaxel was shown to bind to SPARC-rich TMEs; MMP-inhibitors prevent the dissemination of basement membranes and vascular structures resulting in reduced tumor cell invasion; Hyaluronidases depolymerize HA and have suggested to make the TME permissive for immune cell infiltration and drug delivery; Prevention of FAK-phosphorylation prevents FAK activation resulting in increased sensitivity to chemotherapy and immunotherapy; Detection of the extradomain B of fibrinogen by immunocytokines has been proposed to facilitate specific antibody delivery of cytokines such as IL-2. (The displayed figure was partially generated with elements obtained from Servier Medical Art https://creativecommons.org/licenses/by/3.0/legalcode).

**Table 1 cancers-13-00146-t001:** ECM proteins with immune modulatory functions in colorectal (CRC), gastric (GC) and pancreatic cancer (PC).

Matrisome Category	Name	Digestive Cancer	Immune Modulatory Function	Secreted By
Collagens		PC [31] GC [58] CRC [66]	M2 macrophage polarization [66] T cell exclusion [58,59] T cell suppression [62]	Stromal fibroblasts [31,185] Tumor cells [31,185]
Fibrous and non-fibrous glycoproteins	Fibronectin	PC [186] GC [187] CRC [188]	Monocyte recruitment [74,78] Monocyte differentiation [76,77]	Stromal fibroblasts [186] Monocytes/Macrophages [189] Tumor cells [189]
	Laminin	PC [190] GC [191] CRC [192]	T cell infiltration/exclusion [93] DC priming [96]	Tumor basement membrane [82,83]
	Elastin	PC [193] CRC [102]	Macrophage activation [102]	Stromal fibroblasts [100]
Matricellular glycoproteins	SPARC/ Osteonectin	PC [124] GC [194] CRC [195]	Macrophage recruitment/location [125] M2 macrophage suppression [127] TGF-β suppression [129] T reg cell suppression [129]	Stromal fibroblasts [124,195] Macrophages/macrophages [131] Tumor cells [196]
	Osteopontin	PC [197] GC [198] CRC [199]	CD8+ T cell suppression [141] Macrophage recruitment/differentiation [142]	Tumor cells [200,201] Macrophages [201]
	Periostin	PC [202] GC [156] CRC [155]	Macrophage recruitment [155] M2 macrophage polarization [156]	Stromal fibroblasts [150,151,152] Tumor cells [152]
	βig-H3/ TGFβi	PC [168,169] GC [203] CRC [165]	T cell suppression [168,170]	Stromal fibroblasts [168,203] Tumor cells [168,203] Macrophages [173]
Proteoglycans	Versican	PC [204] GC [205] CRC [176,206]	Macrophage differentiation [176] Macrophages- and DC-mediated T cell activation [178,179] T reg cell recruitment [181]	Stromal fibroblasts [174,207] Tumor cells [174,207] Infiltrating leukocytes [174,207]

## Data Availability

No new data were created or analyzed in this study. Data sharing is not applicable to this article.

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
