# Peer review of "Stromal Protein-Mediated Immune Regulation in Digestive Cancers"

_cancers, 2021, doi:10.3390/cancers13010146_

Round 1

Reviewer 1 Report

This manuscript is well written with good flow and use of the English language. It was pleasant to read and summarize a very interesting field. It describes the components of the extracellular matrix of gastric track in relation to the immune response in the first part. To the second part, it summarizes the oncoming therapeutic approaches targeting ECM in the affirmations cancers. I have only one major comment and some small.

My only major comment is that authors use extended abbreviations that make the text unreadable. This is an old way of writing that had to do with strict rules related to world number and space when the paper had to be printed. In the digital era, I strongly recommend using as less as possible abbreviations. For instance, don't write GC but gastric cancer. An example is the first sentences on page 1, lines 34-40. And for this reason, the whole introduction was very hard to follow.

Minor comments

  1. Page 2 line 74-75. Uncomprehensive sentence please check.
  2. Page 8 line 387. Is this a mistake or this is the name of this section?
  3. Table 1 be sure that the table is well structured. Especially the 2 first columns.
  4. 2 interesting papers that are not included and maybe is in authors interest.  https://www.frontiersin.org/articles/10.3389/fimmu.2020.558169/full and https://onlinelibrary.wiley.com/doi/10.1002/ijc.32945. Full disclosure I am not the author in any of those papers.

Author Response

Response to Reviewer 1 Comments

We would like to thank the reviewer for dedicating the time to provide valuable feedback on our manuscript. We are grateful for the insightful comments that helped to improve the quality of our review paper.

Point 1: My only major comment is that authors use extended abbreviations that make the text unreadable. This is an old way of writing that had to do with strict rules related to world number and space when the paper had to be printed. In the digital era, I strongly recommend using as less as possible abbreviations. For instance, don't write GC but gastric cancer. An example is the first sentences on page 1, lines 34-40. And for this reason, the whole introduction was very hard to follow.

Response 1: We exchanged the abbreviation CRC, GC and PC for colorectal cancer, gastric cancer and pancreatic cancer, respectively. Moreover, we exchanged other abbreviations throughout the manuscript to make improve the readability.

Point 2: Page 2 line 74-75. Uncomprehensive sentence please check.

Response 2: We restructured the sentence for more clarity.

Point 3: Page 8 line 387. Is this a mistake or this is the name of this section?

Response 3: It is the name of the subsection. We changed it to SPARC/Osteonection for better understanding and to be coherent with the Table 1.

Point 4: Table 1 be sure that the table is well structured. Especially the 2 first columns.

Response 4: We readjusted the table and we will pay attention to the table format for the resubmission.

Point 5: 2 interesting papers that are not included and maybe is in authors interest.  https://www.frontiersin.org/articles/10.3389/fimmu.2020.558169/full and https://onlinelibrary.wiley.com/doi/10.1002/ijc.32945. Full disclosure I am not the author in any of those papers.

Response 5: We included both references in the section 3.

Reviewer 2 Report

Major comments

The authors provided a wide overview on the modulatory role exerted by stromal proteins in cancers. A  relevant portion of the manuscript  is not clear (i.e. 3.3.1 SPARC paragraph) due to the great number of listed  results that, in some cases, have been poorly described and discussed.  Since the manuscript  should be  focused on digestive cancers, some results might be eliminated or discussed mainly in the context of digestive cancers, with the aim to clarify some concepts. Conversely other results related to digestive cancers should be more deeply and critically discussed.

The discussion about the translational potential of some approaches targeting stromal proteins should be integrated in the various sections containing the description of the proteins  rather than to be inserted  in a  dedicated section  (section 4).

Line 100. Immune cell populations involved in cancer immune-surveillance are not all derived from the thymus! In line with this observation effect of matrisoma on innate NK cell compartment is completely missing. At least a mention of TGF-b, protein activated by ECM (Worthington J.J. et al. TGFb: a sleeping giant awoken by integrins, Trends in biochemical Sciences) and able to  deeply suppress Natural Killer cell activity  should be included  almost in the discussion covered by line 295-303.

Line 176-177. This sentence is misleading… Education? Do the authors mean polarization?

Line 190-196. These sentences are not clear.

3.2.1 Fibronectin section. Fibronectin favors migration of macrophages (ref 70) and by the other side hampers migration of T cells by entrapping them in fibronectin- and collagen rich. As conclusion of this paragraph the authors should better comment these different effects. In case of macrophages the concomitant production of MMP might favor tumor invasion, differently from that occurs in T cells? Please add a comment.

Lines 308-312. The sentence is not clear.

Line 360-362. “Periostin is physiologically expressed in the periosteum, which is a specialized membrane covering the outer surface of bone [147]. Moreover, periostin is expressed in connective tissues that are rich in collagen,….” Periosteum is histologically classified as a connective tissue rich in collagen! Please modify the sentences.

Line 387. Typing error.

Line 388. TGFbi is misleading. Change with TGFb-induced protein.

Line 453-455. The sentence is not clear

Line 464.   4.1. Albumin binding-based agent delivery. I would eliminate this part, albumin is not a target but rather a  tool to deliver...? (see the title of the paragraph).

Line 483. “…poor pharmaceutical characteristics…”. Please, explain the concept more clearly

Minor comments:

Line 1. “…immunosuppression…” is reductive in relation to the  whole content of the review, “modulation” might be  more appropriate.

Line 58. Inflammation and angiogenesis can not be defined  stromal cell functions. Please change the sentence…Es:   ”…by affecting  stromal-regulated  processes such as…

Line 271: 40))

The following reference should be added  and discussed 

Blockade of Stromal Gas6 Alters Cancer Cell Plasticity, Activates NK Cells, and Inhibits Pancreatic Cancer Metastasis. Ireland L. et al, Front. In Immunol 2020 ; PMID: 32174917 DOI: 10.3389/fimmu.2020.00297

Author Response

Response to Reviewer 2 Comments

We would like to thank the reviewer for dedicating the time to provide valuable feedback on our manuscript. We are grateful for the insightful comments that helped to improve the quality of our review paper.

Major comments

Point 1: The authors provided a wide overview on the modulatory role exerted by stromal proteins in cancers. A  relevant portion of the manuscript  is not clear (i.e. 3.3.1 SPARC paragraph) due to the great number of listed  results that, in some cases, have been poorly described and discussed.  Since the manuscript  should be  focused on digestive cancers, some results might be eliminated or discussed mainly in the context of digestive cancers, with the aim to clarify some concepts. Conversely other results related to digestive cancers should be more deeply and critically discussed.

Response 1: We adapted the text and removed redundant examples given for other cancers especially in the sections addressing the different matricellular proteins. However, we think that some of the results obtained in other cancers can be translated to digestive cancers and we aimed to stimulate complementary research approaches for digestive cancers.

Point 2: The discussion about the translational potential of some approaches targeting stromal proteins should be integrated in the various sections containing the description of the proteins  rather than to be inserted  in a  dedicated section  (section 4).

Response 2: We understand the reviewer’s point but we think that an integration is not necessary, since it might overcharge the respective subsection. Moreover, since all the discussed approaches underwent clinical trial, we believe that a dedicated chapter points out already the translational benefit of some of the proteins, while for other targets there is only data available in mice extrapolating a potential therapeutic benefit.

Point 3: Line 100. Immune cell populations involved in cancer immune-surveillance are not all derived from the thymus! In line with this observation effect of matrisoma on innate NK cell compartment is completely missing. At least a mention of TGF-b, protein activated by ECM (Worthington J.J. et al. TGFb: a sleeping giant awoken by integrins, Trends in biochemical Sciences) and able to  deeply suppress Natural Killer cell activity  should be included  almost in the discussion covered by line 295-303.

Response 3: We agree with the reviewer that TGF-b plays an important role in the tumor microenvironment and is especially connected to the processing via the integrins. Nevertheless, TGF-b is not a matrisome protein so we made the choice to not discuss in the review since there is already substantial literature on the subject.

Point 4: Line 176-177. This sentence is misleading… Education? Do the authors mean polarization?

Response 4: In the context of macrophage polarization, the term “education” might be a bit unconventional but it has been used before by other authors (see below). However, for better understanding, we changed it into polarization.

  1. Conejo-Garcia JR, Rodriguez PC: c-Maf: a bad influence in the education of macrophages. J Clin Invest 2020, 130:1629-1631.
  2. Sousa S, Brion R, Lintunen M, Kronqvist P, Sandholm J, Monkkonen J, Kellokumpu-Lehtinen PL, Lauttia S, Tynninen O, Joensuu H, et al.: Human breast cancer cells educate macrophages toward the M2 activation status. Breast Cancer Res 2015, 17:101.
  3. Wu Y, Zheng L: Dynamic education of macrophages in different areas of human tumors. Cancer Microenviron 2012, 5:195-201.

Point 5: Line 190-196. These sentences are not clear.

Response 5: We restructured the sentences for more clarity.

Point 6: 3.2.1 Fibronectin section. Fibronectin favors migration of macrophages (ref 70) and by the other side hampers migration of T cells by entrapping them in fibronectin- and collagen rich. As conclusion of this paragraph the authors should better comment these different effects. In case of macrophages the concomitant production of MMP might favor tumor invasion, differently from that occurs in T cells? Please add a comment.

Response 6: We added a comment emphasising the role of T cell entrapment on the immune evasion of tumor cells. Also, we removed a redundant example for T cell entrapment from that paragraph.

Point 7: Lines 308-312. The sentence is not clear.

Response 7: We restructured the sentences for more clarity.

Point 8: Line 360-362. “Periostin is physiologically expressed in the periosteum, which is a specialized membrane covering the outer surface of bone [147]. Moreover, periostin is expressed in connective tissues that are rich in collagen,….” Periosteum is histologically classified as a connective tissue rich in collagen! Please modify the sentences.

Response 8: We modified the sentences.

Point 9: Line 387. Typing error.

Response 9: We corrected the typing error and adapted the title of the subsection for more clarity.

Point 10: Line 388. TGFbi is misleading. Change with TGFb-induced protein.

Response 10: We added TGF-b-induced protein in addition to TGFbi. Since we are aware of the multiple ways of naming this protein and to avoid more confusion, we referred to it as big-h3 in the text.

Point 11: Line 453-455. The sentence is not clear

Response 11: We restructured the sentence for more clarity.

Point 12: Line 464.   4.1. Albumin binding-based agent delivery. I would eliminate this part, albumin is not a target but rather a  tool to deliver...? (see the title of the paragraph).

Response 12: We changed the title of this particular subsection and moved it to the end of the overall section. However, we decided to keep it in the manuscript since the concept of agent delivery (even through as carrier as albumin) towards a region rich of certain ECM proteins seems to be translational to other treatments.

Point 13: Line 483. “…poor pharmaceutical characteristics…”. Please, explain the concept more clearly

Response 13: We adapted that part for more clarity.

Minnor comments:

Point 14: Line 1. “…immunosuppression…” is reductive in relation to the  whole content of the review, “modulation” might be  more appropriate.

Response 13: We assume that the reviewer refers here to line 16 instead of line 1. In this case, we find the term immunosuppression appropriate since it is referring to the capacity of the TME to dampen immune responses.

Point 15: Line 58. Inflammation and angiogenesis can not be defined  stromal cell functions. Please change the sentence…Es:   ”…by affecting  stromal-regulated  processes such as…

Response 15: We modified the sentence.

Point 16: Line 271: 40))

Response 16: We removed the redundant bracket.

Point 17: The following reference should be added  and discussed 

Blockade of Stromal Gas6 Alters Cancer Cell Plasticity, Activates NK Cells, and Inhibits Pancreatic Cancer Metastasis. Ireland L. et al, Front. In Immunol 2020 ; PMID: 32174917 DOI: 10.3389/fimmu.2020.00297

Response 17: We included the references in the section 3.

Reviewer 3 Report

In this review article, the authors nicely and comprehensively summarized the published studies investigating the roles of the tumor microenvironment and stromal factors regulating digestive malignancies. Overall, the manuscript is well written and is a useful contribution to the literature particularly for teaching purposes. There are a few points, which must be addressed.

1) Title of the manuscript is not in line with the text. In many sections of the review authors extensively talk about other cancers including breast, ovarian, brain, etc, which is not what the title suggests. All of these examples must be removed and the text must be focused only on digestive cancers. Also, I would recommend changing the title as well. Stromal-protein mediated is a problematic statement.

2) Apart from summarizing the current literature, authors should also discuss what future research should be done in the field. This is only done with a few sentences. It will be helpful to guide other people interested in this field.

3) The term “immune cytokines” in text and figures should be replaced with just “cytokines”. Many cytokines are also secreted by non-immune cells.

Author Response

Response to Reviewer 3 Comments

We would like to thank the reviewer for dedicating the time to provide valuable feedback on our manuscript. We are grateful for the insightful comments that helped to improve the quality of our review paper.

Point 1: Title of the manuscript is not in line with the text. In many sections of the review authors extensively talk about other cancers including breast, ovarian, brain, etc, which is not what the title suggests. All of these examples must be removed and the text must be focused only on digestive cancers. Also, I would recommend changing the title as well. Stromal-protein mediated is a problematic statement.

Response 1: We adapted the text and removed some of the somewhat redundant examples given for other cancers. However, we think that some of the results obtained in other cancers can be translated to digestive cancers and by discussing the corresponding literature we aimed to stimulate complementary research approaches.

We do understand that stromal-protein mediated might have a semantic problem. However, we believe that we give a new perspective on the fact that secreted proteins in the extracellular matrix from the cells located in the stroma are interacting primarily with the immune system and therefore impact tumorigenesis. We do believe that these interactions are key elements (with the support of the overview of the literature) driving the outcome of the solid cancer.

Point 2:  Apart from summarizing the current literature, authors should also discuss what future research should be done in the field. This is only done with a few sentences. It will be helpful to guide other people interested in this field.

Response 2: We included a perspective into the conclusion part.

Point 2: The term “immune cytokines” in text and figures should be replaced with just “cytokines”. Many cytokines are also secreted by non-immune cells.

Response 3: We changed “Immune cytokines” into “Immunocytokines” as in the original article Here, immunocytokines specifically revers to the generated complex of IL-2 and an antibody directed against the extradomain B of fibronectin.

Reference: Wagner K, Schulz P, Scholz A, Wiedenmann B, Menrad A: The targeted immunocytokine L19-IL2 efficiently inhibits the growth of orthotopic pancreatic cancer. Clin Cancer Res 2008, 14:4951-4960.

Round 2

Reviewer 2 Report

Dear authors, 

please add acronym for epidermal GFR at line 42.

Best Regards